# Probabilistic Risk Assessment in Space Launches Using Bayesian Network with Fuzzy Method

Xing Pan [1], Song Ding [1], Wenjin Zhang [1,*], Tun Liu [1], Liqin Wang [2] and Lijing Wang [3]

1   School of Reliability and Systems Engineering, Beihang University, Beijing 100191, China;
    panxing@buaa.edu.cn (X.P.); dingsong98@buaa.edu.cn (S.D.); liu_tun@163.com (T.L.)
2   Beijing Special Engineering and Design Institute, Beijing 100082, China; 18612449107@163.com
3   School of Aeronautic Science and Engineering, Beihang University, Beijing 100191, China;
    wanglijing@buaa.edu.cn
*   Correspondence: author: zwjok@buaa.edu.cn

**Abstract:** Space launch projects are extremely risky, and any equipment failure or human error may lead to disastrous consequences. Probabilistic risk assessment (PRA) is beneficial to qualitative analysis of risk, but it has not been paid enough attention in risk analysis for space launch systems (SLSs). Compared with most qualitative risk analysis in this field, this paper proposes a risk analysis framework based on Bayesian network (BN) with fuzzy method, which is suitable for probabilistic risk analysis of SLS. This method establishes a risk analysis model of SLS based on statistics and expert experience and reduces the uncertainty of the model by using fuzzy theory. By predicting the system risk probabilities, diagnosing the key risk causes, determining the risk conduction path, and performing a sensitivity analysis, the proposed risk analysis framework is aimed at alleviating this drawback to deal more effectively with the uncertainties in the field of space launches. A case study of space launches demonstrates and verifies the proposed method, and it also provides guidance for similar engineering projects.

**Keywords:** probability risk assessment; Bayesian network; fuzzy method; space launch system



## 1. Introduction

With the development of science and technology, the space launch technology of many countries has become increasingly mature, and the risk of space launches has thus been gradually maintained under strict control [1]. However, in recent years, several major accidents in space launch missions have occurred worldwide and these critical launch accidents show that considerable risks pertaining to the SLS still exists, which also means that further research on the risk analysis of the SLS needs to be performed [2,3]. The statistics in China show that a launch centre has successfully completed 47 launch missions in a certain past decade, which included nearly 200 potential launch risks. The causes of these potential launch risks include design defects, improper operation, software defects, organisation and command issues, and system interference. The above space launch accidents and statistical analyses show that even if a launch mission is successfully completed, there still exist many potential risks in the launch processes. It is thus crucial to assess the risk of the SLS before launch.

At present, the PRA is mainly used to quantitatively analyse the system risks in high-risk engineering fields. In general, PRA refers to a new class of system risk analyses and accident evaluation methods developed by the U.S. in the field of nuclear power after the 1960s. This approach mainly adopts system reliability evaluation technologies, such as fault tree analysis and event tree analysis to comprehensively analyse the occurrence and process of possible accidents in complex systems and take into account the occurrence probability and consequences of such accidents [4]. The main purpose of these methods in the system risk analysis is to analyse and evaluate the probability of dangerous events, accidents, or

failures [5,6]. At present, a large number of PRA methods have been developed for the fields of nuclear power, aerospace, and maritime transportation, etc. [7].

Based on the above PRA methods, many scholars have analysed the risk of in space launch. A risk analysis method based on the information fusion is proposed to assess the mean collective risk to the general public after a rocket launch, which provides a reference for the risk aversion after a rocket launch [8]. The impact of environmental factors on the safety risk of a space launch is studied through the time series analysis of a large amount of space launch data [9]. The systems-theoretic process analysis is applied to an SLS to improve the safety of rocket launch events [10]. It is proposed to strengthen the safety risk assessment of the rocket launch and re-entry events and construct a framework for the risk assessment [11]. The safety risks of space launch events in Australia are researched and a method to avoid the risks and improve the system safety is proposed [12]. A risk management model is proposed to attempt to establish risk management standards to standardise the space launch missions of various countries [13]. All the studies mentioned extend the application of PRA to the risk assessment of rocket launch processes. However, there are still some limitations. Some studies assign precise probabilities to the basic risk events, which will inevitably introduce uncertainty to a certain extent due to the scarcity of data and lack of knowledge, as this does not satisfy the law of large numbers in probability theory [14].

Fuzzy set theory, which can also be called fuzzy method, was proposed by Zadeh in 1965 [15]. It is an extension of the classical set theory. In fuzzy sets, the relationship between elements and sets is no longer an absolute state of belonging and not belonging, instead, it is represented by a membership function with the interval between [0, 1]. It is regarded as an effective tool to deal with the cognitive uncertainty brought by small sample data [16]. Some researchers apply fuzzy theory to QRA by combining it with event tree analysis (ETA) and fault tree (FTA) analysis [17,18]. A new approach combining fuzzy theory and the HAZOP technique is proposed with application in the risk analysis of gas wellhead facilities [19]. Nevertheless, the traditional PRA method has great challenges in handling fuzzy data under uncertainty conditions [20]. For instance, most of the existing studies only assessed the impact of the risk factors on the whole system and only a few of them considered the causal relationship among the risk factors, and the tree-based model uses Boolean logic, which limits the validity of the causal probability of the model.

A Bayesian network, also known as belief network, is a directed acyclic graph. It combines graph theory and probability theory and can be expressed as <V, E>, where V denotes the nodes and E denotes the directed edges between nodes. In a BN, nodes represent random variables, and directed edges between nodes represent the causal relations between nodes (from the parent node to its children), expressing the strength of the relations with conditional probability tables (CPTs), and prior probabilities are used to present those without parents [21]. A BN is suitable for the expression and analyses of uncertainties or probabilistic events since it can generate reasoning from uncertain knowledge or information [22]. It is considered as a robust risk analysis technique since it can represent causal relationships among events and perform inference of risk events with new evidence under uncertainty. Therefore, this approach is widely used in the reliability assessments [23], fault diagnoses [24], and failure probability upgrades of safety systems [25]. The BN graphically shows the causal relationship among the variables and uses the probability to quantitatively clarify the development of these causal relationships; subsequently, the network diagnoses the problems, calculates the posterior probability of the variables after obtaining new information (evidence), and updates the model. This approach can support forward reasoning, sensitivity analysis, and backward reasoning, which enables it to realise the risk management of the entire process that includes prior prediction, construction control, and subsequent diagnosis. A prior research combined the interpretive structural model (ISM) and BN to quantitatively analyse the relationships and interaction strengths between risk factors in the railroad hazardous goods transportation system [26]. Yin and Li applied BN to quantitative risk analysis of offshore well blowout

accidents [27]. An approach mapping fault tree to BN is proposed to analyse the risk of ship grounding accidents. Although BN is used in many fields, based on the high-risk nature of space launches, it is necessary to combine fuzzy theory and applying expert experience and historical statistics to perform risk analysis of space engineering.

In view of the current limitations of the process risk analysis in space launches, this paper proposes a risk analysis framework suitable for the process of SLSs based on BNs, which uses the fuzzy theory to reduce the uncertainties of insufficient statistics expert evaluation. The rest of this paper is organized as follows. Section 2 introduces the methodology including the mathematical basis, modelling process, and the fuzzy method of BNs. Section 3 shows a case study of the process risk analysis for SLSs including case background, causal structure, and probabilities of the BN. Section 4 gives the risks pertaining to the SLS including forward reasoning, reverse reasoning, and sensitivity. Some issues are also discussed in this section. Section 5 summarises the paper.

## 2. Methodology

### 2.1. Application Framework of BNs

When using the BN to analyse the process pertaining to an SLS, the causal model of the system should be constructed first according to the causal logic of the risk events or accidents; subsequently, the prior probability of the node events and conditional probability among the nodes can be assigned through multiple data sources. The application of BNs for risk analysis is mainly divided into the following three steps: constructing the causal logic model of the system risk; constructing the CPTs among nodes in the BN; and applying the BN to process risk analysis.

Combining the universal application framework of the BN method and the requirement of safety analysis for the SLS, this paper proposes a process risk analysis framework based on BNs, as shown in Figure 1. The new framework is divided into three levels, including principle, model, and application. The principle layer is the underlying principle of constructing a BN model, including history statistics, expert experience, and fuzzy theory. Because it is difficult to obtain a large number of historical data, it needs expert evaluation data to supplement, and fuzzy theory can reduce the uncertainty of subjective evaluation. A complete BN model must include causal model, prior probability, and CPT, which can be obtained by the combination of the principles of the principle layer. The application layer contains some application analysis, which are carried out after the BN model is obtained.

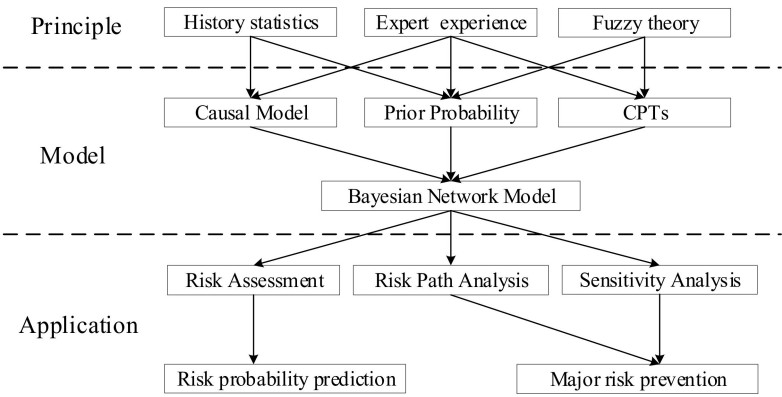

**Figure 1.** BN application framework for the risk analysis.

### 2.2. Mathematical Basis of BNs

BNs, also known as belief networks or directed acyclic graphs, are probability graph models that contain several nodes and arrows. The nodes represent the random variables and arrows represent the dependencies or causal relationships among the variables. Accord-

ing to the conditional independence and chain rule [21], the joint probability distribution of a set of variables $U = \{A_1, A_2, \ldots, A_n\}$ can be determined using

$$P(U) = \prod_{i=1}^{n} P(A_i | Pa(A_i)) \tag{1}$$

where $Pa(A_i)$ denotes the "cause" of $A_i$ or the parents of $A_i$ in the BN. The solution of the conditional probability in the BN depends on the Bayesian theorem, as shown in Formula (2). Given the observation of another set of variables $E$ called the evidence, the posterior probability distribution of a particular variable can be computed by using different classes of inference algorithms, such as the junction tree or variable elimination based on the Bayesian theorem as follows:

$$P(U|E) = \frac{P(E|U)P(U)}{P(E)} = \frac{P(E, U)}{\sum_U P(E, U)} \tag{2}$$

*2.3. Fuzzification and Defuzzification*

2.3.1. Fuzzy BNs

Fuzzy BNs (FBNs) are obtained when fuzzy theory is applied to BNs. Zadeh first proposed that membership functions can be used to well simulate the vagueness in experts' evaluation [28], so as to replace the representation of precise probability distributions [29]. However, the membership functions cannot be directly applied to BNs [30], because the fuzzy measure does not satisfy the rule of probability [31]. Using fuzzy probability methods [32,33] and using the conversion between fuzzy measure and probability measure [34] are two feasible solutions, which enable fuzzy measure to be widely used in BNs [35,36]. In FBNs, expert elicitation is inevitably involved, no matter how, to obtain required fuzzy probabilities. Expert elicitation is applied in calculating the probabilities of vague events [30] and is a solution for dealing with uncertainties as well as a lack of sufficient data, providing useful information for risk analysis [37]. Language variables are very effective in dealing with ambiguous or poorly defined situations, and some vague language ranges given in advance can also reduce the uncertainty of the assessment to some extent [15]. For the sake of expert elicitation, this paper also designs the corresponding prior probabilities and CPTs of the proposed method, so that experts can give a score by evaluating fuzzy language variables, which could be converted into fuzzy probabilities quickly and conveniently.

2.3.2. Fuzzification of Prior Probabilities

This paper used a fuzzy method to fuzzify the state of risk nodes in the system, and the state of nodes is defined in the form of three risk levels, namely, "High risk", "Medium risk", and "Low risk". In order to transform two state events into multi-state events, risk events of the same type are formed into a set of risk events with the risk probability interval of $[i, j]$. Domain experts need to use fuzzy theory to give the evaluation values of parameters $\theta_1$ and $\theta_2$, based on which the probability intervals can be obtained. These are shown in Table 1. Figure 2 shows the fuzzy membership range of three risk levels and their probability intervals, which the risk level is determined according to. Let $j - i = m$, $p$ as the probability of events, then the fuzzy trigonometric function in Figure 2 is shown in Formulas (3)–(5).

**Table 1.** Node state and its corresponding probability interval based on fuzzy trigonometric function.

| Node State | Probability Interval of the Risk Levels |
|:---:|:---:|
| Low risk | $\left[i, i + \frac{\theta_1 + \theta_2}{2}(j - i)\right]$ |
| Medium risk | $[i + \theta_1(j - i), i + \theta_2(j - i)]$ |
| High risk | $\left[i + \frac{\theta_1 + \theta_2}{2}(j - i), j\right]$ |

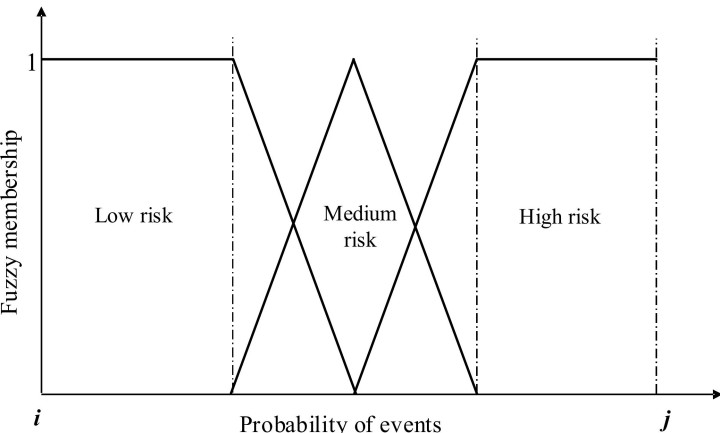

**Figure 2.** Mapping between the fuzzy membership and probability of the events.

$$F_{low\ risk}(p) = \begin{cases} 1 & ,i \leq p \leq i + \theta_1 m \\ \frac{-2p+(\theta_1+\theta_2)m}{-2i+(\theta_2-\theta_1)m}, & i + \theta_1 m < p \leq i + \frac{\theta_1+\theta_2}{2}m \end{cases} \tag{3}$$

$$F_{medium\ risk}(p) = \begin{cases} \frac{2p-2(i+\theta_1 m)}{-2i+(\theta_2-\theta_1)m}, i + \theta_1 m \leq p \leq i + \frac{\theta_1+\theta_2}{2}m \\ \frac{-2p+2(i+\theta_2 m)}{2i+(\theta_2-\theta_1)m}, i + \frac{\theta_1+\theta_2}{2}m < p \leq i + \theta_2 m \end{cases} \tag{4}$$

$$F_{high\ risk}(p) = \begin{cases} \frac{2p-(\theta_1+\theta_2)m}{2i+(\theta_2-\theta_1)m}, i + \frac{\theta_1+\theta_2}{2}m \leq p < i + \theta_2 m \\ 1 & ,i + \theta_2 m \leq p \leq j \end{cases} \tag{5}$$

Through the above transformation, the historical statistical data represented by probability value can be transformed into the fuzzy membership of three-level risk. However, what is calculated in BNs is the probability, so it is necessary to transform the fuzzy membership into fuzzy probability [38]. For the triangular fuzzy membership function used above, the fuzzy membership degree and fuzzy probability are uniform on [0, 1]. Therefore, what needs to be considered is to normalise the sum of subordinate degrees. Due to the particularity of the constructed membership function, $\sum F_{risk}(p)$, which is the sum of membership degree for any probability $p$, constantly equals 1, meeting the requirements of fuzzy probability, as shown in Formula (6). The fuzzy probability at three risk levels is shown in the Formulas (7)–(9).

$$\sum F_{risk}(p) = \sum P(risk) = 1 \ \forall p \in [i, j] \tag{6}$$

$$P(risk = low) = \frac{F_{low\ risk}(p)}{\sum F_{risk}(p)} = F_{low\ risk}(p) \tag{7}$$

$$P(risk = medium) = \frac{F_{medium\ risk}(p)}{\sum F_{risk}(p)} = F_{medium\ risk}(p) \tag{8}$$

$$P(risk = high) = \frac{F_{high\ risk}(p)}{\sum F_{risk}(p)} = F_{high\ risk}(p) \tag{9}$$

### 2.3.3. Fuzzification of CPTs

To determine the conditional probability between the parent node and the child node of the causal network constructed, the expert evaluation method is combined with the fuzzy evaluation method to generate objective and effective CPTs by relying less on the expert precise evaluation data and to reduce the uncertainties of the CPTs. If the states of all nodes are set to three states of high, medium, and low risk, it only needs to evaluate the risk states corresponding to the children nodes when each parent node is in a certain risk state. The triangular fuzzy function is used to establish a scoring framework of the

risk level using the evaluation of domain experts, as shown in Figure 3. Consequently, the influencing weight of the parent node on the child node and the state of the parent node are comprehensively considered to calculate the probability of the child node in each state.

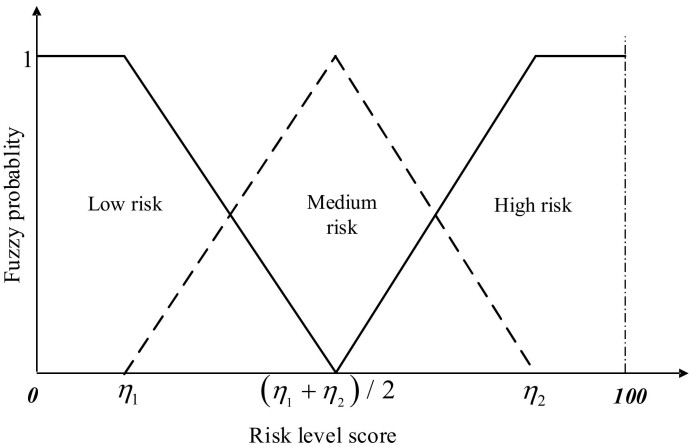

**Figure 3.** Scoring framework of the risk level.

Firstly, a scoring framework is established to measure the risk level. In Figure 3, the functions between the probability of the risk level (low risk, medium risk, high risk) and the score of the risk level are $f(x)$, $g(x)$, and $h(x)$, respectively. The specific function expressions are shown in Formulas (10)–(12), respectively.

$$f(x) = \begin{cases} 1, & 0 \le x \le \eta_1 \\ \frac{-2x+(\eta_1+\eta_2)}{\eta_2-\eta_1}, & \eta_1 < x \le \frac{\eta_1+\eta_2}{2} \end{cases} \tag{10}$$

$$g(x) = \begin{cases} \frac{2x-2\eta_1}{\eta_2-\eta_1}, & \eta_1 \le x \le \frac{\eta_1+\eta_2}{2} \\ \frac{-2x+2\eta_2}{\eta_2-\eta_1}, & \frac{\eta_1+\eta_2}{2} < x \le \eta_2 \end{cases} \tag{11}$$

$$h(x) = \begin{cases} \frac{2x-(\eta_1+\eta_2)}{\eta_2-\eta_1}, & \frac{\eta_1+\eta_2}{2} \le x < \eta_2 \\ 1, & \eta_2 \le x \le 1 \end{cases} \tag{12}$$

Then, the corresponding score is obtained by using the risk level, which can be determined by the mean value of the risk score. According to the above constructed trigonometric function chart and the functional formula, the mean value of the risk probability corresponding to the high, medium, and low risks in the [0, 1] range can be obtained as follows:

$$E(P = low\ risk) = \frac{1}{\eta_1}\int_0^{\eta_1} x dx - \frac{4}{(\eta_2-\eta_1)^2}\int_{\eta_1}^{\frac{\eta_1+\eta_2}{2}} \left(x - \frac{\eta_1+\eta_2}{2}\right) x dx = \frac{11\eta_1+\eta_2}{12} = E_1 \tag{13}$$

$$E(P = medium\ risk) = \frac{\eta_1+\eta_2}{2} = E_2 \tag{14}$$

$$E(P = high\ risk) = \frac{1}{\eta_2}\int_0^{\eta_2} x dx + \frac{4}{(\eta_2-\eta_1)^2}\int_{\frac{\eta_1+\eta_2}{2}}^{\eta_2} \left(x - \frac{\eta_1+\eta_2}{2}\right) x dx = \frac{\eta_1+11\eta_2}{12} = E_3 \tag{15}$$

Assuming that a child node has $n$ parent nodes, then there are $3^n$ conditional probability combinations composed of $n$ parent nodes. In order to construct the CPTs of the child nodes corresponding to the $3^n$ situations, the expert evaluates the influence weight of the

risk state of the *i* parent node on the child node as $w_i$ ($\Sigma w_i = 1$). Then, the risk score of the child node can be obtained:

$$E_{child} = \sum_{i=1}^{n} w_i \times E_j \, , j \in \{1,2,3\} \tag{16}$$

Therefore, the probability of the child node in each risk level can be expressed as follows:

$$P_{\text{subnode}}(risk = low) = f(E_{child}) \tag{17}$$

$$P_{\text{subnode}}(risk = medium) = g(E_{child}) \tag{18}$$

$$P_{\text{subnode}}(risk = high) = h(E_{child}) \tag{19}$$

The probability of each risk level of each sub node under each combination can be calculated by the above formulas, which forms a complete CPT. Firstly, the risk level is determined by the risk score. Then, the weights of the parent node on the child node are evaluated. Finally, the risk levels of sub nodes are determined by the score calculated by weight to obtain the CPTs.

2.3.4. Defuzzification

With the help of the probabilistic reasoning of the BN, the probability of all of the nodes in the BN at the three risk levels (high risk, medium risk, low risk) could be calculated. However, to further obtain the occurrence probability of each node, the defuzzification process needs to be performed. According to the triangular fuzzy function shown in Figure 2 and the defuzzification Formula (20) [39], the risk probability of each node in the BN could be calculated. By this formula, the defuzzification of fuzzy BNs can be obtained, and the risk evaluation of the BNs can be transformed into the probability measure.

$$\text{Risk probability} = \frac{\int_x x m(x) dx}{\int_x m(x) dx} \tag{20}$$

where $m(x)$ is the overall mapping function between the probability of the risk level and the probability of the events shown in Figure 2.

**3. Case Study**

*3.1. Case Background*

The case background is the fuel filling process of liquid oxidant and liquid reducing agent in space launch sites, using a low temperature filling system and needing to be paid enough attention [40,41]. Both the U.S. and Europe have specialised personnel and institutions for the safety management of low-temperature filling. In contrast, China's space launch sites still need personnel to complete many important operations due to the relatively low degree of automation, so there are many unavoidable safety risks [40]. Liquid hydrogen and oxygen have been widely used as fuel for large launch vehicles [42]. Due to the high risk in the process of low-temperature refutation and the reliance on expert evaluation, many cases need to rely on experts with a right to speak in this field for evaluation and judgment [43]. Since a large amount of subjective evaluation is used in the fuel filling process, it is appropriate to use the BN method in this case. Moreover, many BN methods and fuzzy theories are also applied in modelling to comprehensively evaluate the filling process at space launch sites [44].

*3.2. Construction of the BN Causal Structure*

The application objective of this paper is the space launch site system, which is a specific area for launching spacecraft, and whose main function is to enable the completion of the assembly, test, and launch of the launch vehicles and spacecraft. A space launch site usually consists of a technical zone (test site), a launch zone (launch site), a launch command

and control centre, and a ground measure-control system (or ground TT&C system). The technical zone is a devoted area for technical preparation, and its main objective is to allow the assembly and testing of the launch vehicles and spacecraft and the testing of the individual instruments and equipment of the internal systems of these vehicles. The launch zone conducts the preparation and launch, and the launch command and control centre commands, monitors, and manages the launch test of the spacecraft. The measure-control system is a set of ground facilities for tracking and measuring the launch vehicles and spacecraft, receiving the telemetry and external measurement information, as well as sending the monitoring data, safety instructions, and communication information [45]. During the launch of launch vehicles, the technical zone, launch zone, and ground measure-control system are mainly oriented to the tasks before launch, while the launch command and control centre is mainly oriented to the tasks after launch. We invited five experts at a Chinese Satellite launch centre to conduct a risk event analysis to structure BN. All the experts have more than 5 years of experience in the field. Experts analysed the mission risk before the launch, and thus mainly considered the risk pertaining to three areas: the technical zone, launch zone, and ground measure-control system.

After statistically analysing 216 accidents that occurred in the Jiuquan Satellite Launch Centre, China Satellite Maritime Tracking and Control Department, Taiyuan Satellite Launch Centre, Xi'an Satellite Control Centre and Xichang Satellite Launch Centre during 2004–2012, combined with the field investigation results of these space launch sites, the authors noted that some key risk events occur frequently or have severe consequences once they occur. Since the frequency and severity of the risk events are two important indicators for the risk assessment, this paper closely analysed 15 key risk events with a high level of severity and assumed that the risk events with a low level of severity do not occur. The specific descriptions of the 15 key risk events are presented in Table 2.

**Table 2.** Key risk events and their descriptions.

| No. | Key Risk Event | Description |
|:---:|:---:|:---:|
| 1 | Frequency multiplier | The output spectrum clutter of the phase-locked frequency multiplier increases. |
| 2 | Front connecting rod | The cable plug of the front connecting rod is damaged by a screw that fell from the tower. |
| 3 | Accelerometer | An abnormality of the accelerometer +Z channel pulse occurs at the flight software test bench. |
| 4 | Control cable socket | The control cable socket of the cutter for the satellite rocket separation is disconnected. |
| 5 | Thrust utilisation computer | The thrust utilization computer is unable to start properly. |
| 6 | Geodetic software | Abnormal exit of the strapdown geodetic software occurs. |
| 7 | Commander's instruction | Wrong, delayed or advanced instruction is issued by the commander. |
| 8 | Ladder truck | A fault of the ladder truck occurs. |
| 9 | Pin of the socket | The pin retraction of the socket leads to an open circuit. |
| 10 | JQ2 regulator | A fault of the JQ2 regulator in the helium distribution pre-launch platform of the cryogenic power system occurs. |
| 11 | Servo mechanism | The servo mechanism is damaged due to improper operation. |
| 12 | Booster compartment | The hatch door of the booster compartment is not closed. |
| 13 | Hard hose plug for filling | The hard hose plug for filling the primary incendiary agent is ejected. |
| 14 | Automobile power station | A power supply interruption is caused by a fire in the automobile power station. |
| 15 | Malindy station | The uplink remote control command of the Malindy station is not sent. |

*3.3. Determination of Probabilities in the BN*

3.3.1. Conversion of Prior Probabilities

According to the historical data of some space launch sites in China, a total of 47 space launch missions have been conducted from 2004 to 2012. This paper divided the total number of key risk events that occurred by the sum of the launch missions, and this frequency is considered as the prior probability of the 15 key risk events. The result is shown in Table 3.

**Table 3.** Prior probability of the key risk events.

| No. | Key Risk Event | Number | Probability |
|---|---|---|---|
| 1 | Frequency multiplier | 1 | 0.0213 |
| 2 | Front connecting rod | 1 | 0.0213 |
| 3 | Accelerometer | 1 | 0.0213 |
| 4 | Control cable socket | 2 | 0.0426 |
| 5 | Thrust utilisation computer | 3 | 0.0638 |
| 6 | Geodetic software | 1 | 0.0213 |
| 7 | Commander's instruction | 4 | 0.0851 |
| 8 | Ladder truck | 2 | 0.0426 |
| 9 | Pin of the socket | 2 | 0.0426 |
| 10 | JQ2 regulator | 1 | 0.0213 |
| 11 | Servo mechanism | 1 | 0.0213 |
| 12 | Booster compartment | 1 | 0.0213 |
| 13 | Hard hose plug for filling | 1 | 0.0213 |
| 14 | Automobile power station | 1 | 0.0213 |
| 15 | Malindy station | 1 | 0.0213 |

Table 3 shows that the probability interval of the key risk events is [0.0213, 0.0851]; thus, this paper preliminarily set the probability interval of the events corresponding to the risk level as [0, 0.1]. According to expert experience, the probability intervals of the risk events corresponding to each risk level are divided as presented in Table 4. The relationship between the probability of the risk level and the probability of the events is mapped by means of the triangular fuzzy function, as shown in Figure 4.

**Table 4.** Probability interval of the risk levels.

| Risk Levels | Low Risk | Medium Risk | High Risk |
|---|---|---|---|
| Probability interval | [0, 0.04] | [0.02, 0.06] | [0.04, 0.1] |

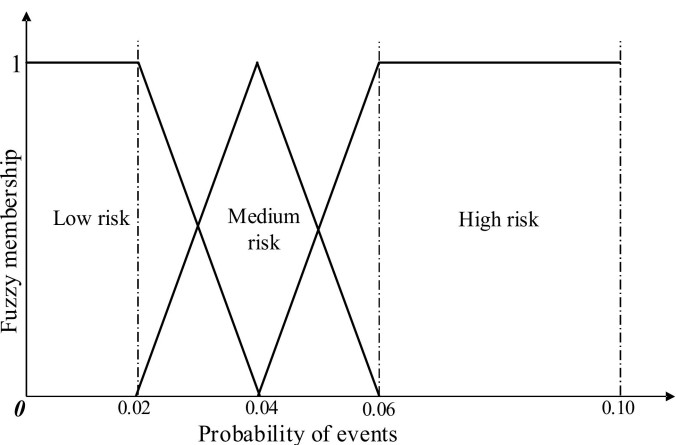

**Figure 4.** A mapping example between risk level and events.

In the risk causal network constructed in Figure 4, all the root nodes are assigned the prior probabilities according to the historical statistical data. Then, the risk probabilities



of the remaining nodes and the conditional probabilities between the parent nodes and child nodes can be fuzzified using the following subjective evaluation method. To facilitate the construction of the CPTs, on the basis of the relationship mapped in Figure 4, the prior probability of the 15 key risk events presented in Table 2 can be converted into the probability of the three risk levels, and the conversion results are presented in Table 5.

**Table 5.** Risk level probability of the key risk events.

| Key Risk Events | 1 | 2 | 3 | 4 | 5 | 6 | 7 | 8 |
|---|---|---|---|---|---|---|---|---|
| P(high risk) | 0 | 0 | 0 | 0.13 | 1 | 0 | 1 | 0.13 |
| P(medium risk) | 0.065 | 0.065 | 0.065 | 0.87 | 0 | 0.065 | 0 | 0.87 |
| P(low risk) | 0.935 | 0.935 | 0.935 | 0 | 0 | 0.935 | 0 | 0 |
| Key risk events | 9 | 10 | 11 | 12 | 13 | 14 | 15 | |
| P(high risk) | 0.13 | 0 | 0 | 0 | 0 | 0 | 0 | |
| P(medium risk) | 0.87 | 0.065 | 0.065 | 0.065 | 0.065 | 0.065 | 0.065 | |
| P(low risk) | 0 | 0.935 | 0.935 | 0.935 | 0.935 | 0.935 | 0.935 | |

### 3.3.2. Subjective Fuzzy Evaluation for CPTs

In this case, the evaluation of CPTs needs experts to use fuzzy theory to carry out subjective evaluation. In order to better show this process, a CPT of certain node is introduced as an example. Taking the CPT construction between the node "Risk in measure-control system" and its parent node as an example, the CPT construction is used to complete the probability data input of Table 6. Since the nodes "14 Automobile power station" and "15 Malindy station" exerted different influences on their child nodes, the influencing weight of the parent node on child node is determined by the expert scoring method, assuming that the weights are $w_1$ and $w_2$, respectively. Three experts with more than ten years of experience in the field conducted the evaluation, and after discussion they came to a consensus result.

**Table 6.** CPT between the "Risk in measure-control system" node and its parent nodes.

| "14 Automobile Power Station" | | High | | | Medium | | | Low | | |
|---|---|---|---|---|---|---|---|---|---|---|
| "15 Malindy Station" | | High | Medium | Low | High | Medium | Low | High | Medium | Low |
| "Risk in measure-control system" | High | 0.833 | 0.342 | 0 | 0.492 | 0 | 0 | 0.15 | 0 | 0 |
| | Medium | 0.167 | 0.658 | 0.85 | 0.508 | 1 | 0.508 | 0.85 | 0.658 | 0.167 |
| | Low | 0 | 0 | 0.15 | 0 | 0 | 0.492 | 0 | 0.342 | 0.833 |

Firstly, experts evaluate this node and give the standard of risk level score. The scoring framework is shown in Figure 5, whose specific function expressions are shown in Formulas (21)–(23), respectively.

$$f(x) = \begin{cases} 1, & (0,20) \\ \frac{50-x}{30}, & (20,50) \end{cases} \tag{21}$$

$$g(x) = \begin{cases} \frac{x-20}{30}, & (20,50) \\ \frac{80-x}{30}, & (50,80) \end{cases} \tag{22}$$

$$h(x) = \begin{cases} \frac{x-50}{30}, & (50,80) \\ 1, & (80,100) \end{cases} \tag{23}$$

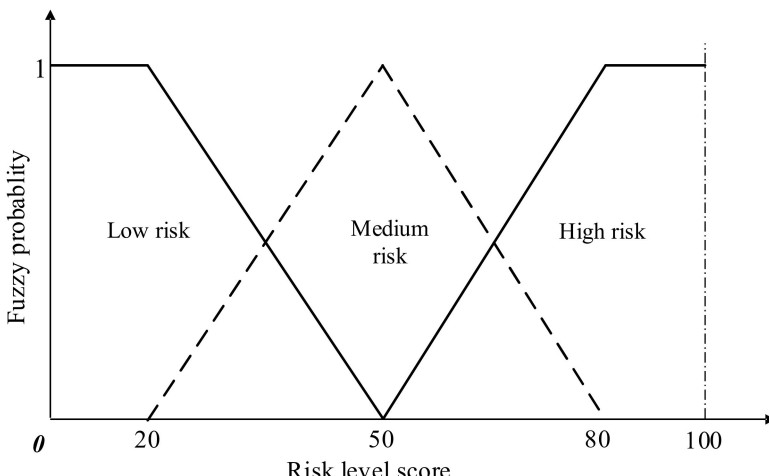

**Figure 5.** A scoring framework example of the risk level.

Experts believe that node "14 Automobile power station" and node "15 Malindy station" have the same impact on the child node "Risk in measure-control system", so both $w_1$ and $w_2$ are 0.5. The score of high/medium/low risk for the two parent nodes is 75/50/25 and then the score of sub node can be obtained. For example, when node "14 Automobile power station" = High and node "15 Malindy station" = High, the risk level probabilities of child nodes are 0.8333/0.1666/0. Similarly, the probability calculation of other combinations can be completed, and the final CPT is shown in Table 6.

According to the abovementioned method, the CPTs among the other child nodes and parent nodes in the causal network can be constructed. This method only requires experts to evaluate the influencing weights of each parent node on the child nodes, and each CPT can be generated through the scoring framework of the risk level shown in Figure 5. The influencing weights of the parent node on the child node, as evaluated by the experts, are listed in Table 7.

**Table 7.** Influencing weights of the parent nodes on the child nodes.

| Parent Node | Weight | Parent Node | Weight |
|---|---|---|---|
| Root node 1 | 0.23 | Root node 12 | 1 |
| Root node 2 | 0.35 | Root node 13 | 1 |
| Root node 3 | 0.18 | Root node 14 | 0.41 |
| Root node 4 | 0.24 | Root node 15 | 0.59 |
| Root node 5 | 0.21 | Process control risk | 0.10 |
| Root node 6 | 0.10 | Rocket stability risk | 0.35 |
| Root node 7 | 0.58 | Launch delay risk | 0.18 |
| Root node 8 | 0.11 | Casualty risk | 0.37 |
| Root node 9 | 0.42 | Risk in technical zone | 0.20 |
| Root node 10 | 0.28 | Risk in launch zone | 0.60 |
| Root node 11 | 0.30 | Risk in measure-control system | 0.20 |

## 4. Results and Discussion

### 4.1. Reasoning Results

According to the abovementioned fuzzy method, combined with the expert evaluation, the authors can obtain the probability of the leaf nodes in the BNs and the conditional probability between the child nodes and parent nodes to obtain the CPTs for the whole network. The CPTs are combined with causal networks to construct the BNs for the space launch risk, as shown in Figure 6.

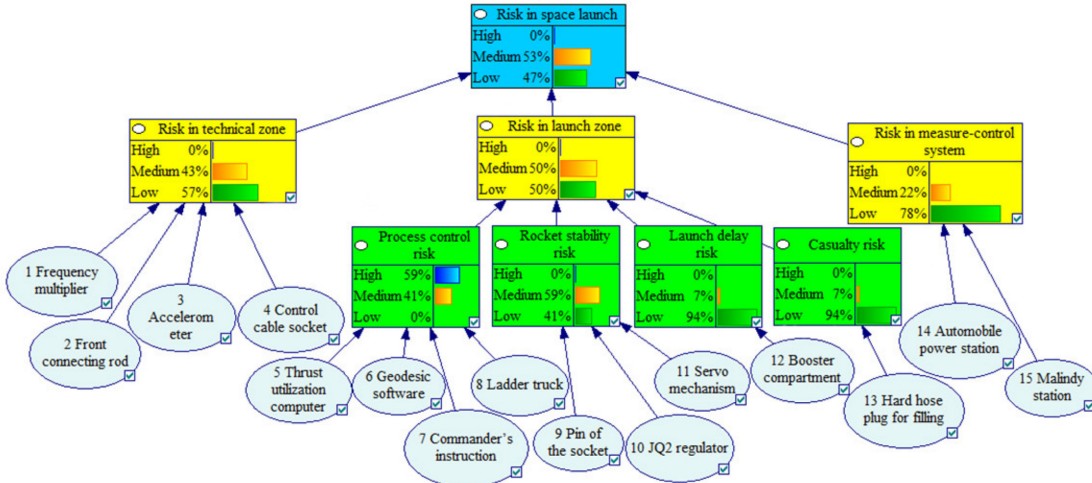

**Figure 6.** BN for the risks in a space launch.

The results of the Bayesian reasoning showed that the probability of the SLS in the high, medium, and low risk states is 0.00004, 0.52683, and 0.47313, respectively, as shown in Figure 6. Finally, the risk probabilities of each node are obtained, as presented in Table 8. The results showed that the total risk probability of the SLS is 0.0306, and the risk in the launch zone is higher than that in the technical zone and ground measure-control system; therefore, the relevant personnel should focus on reducing the risk in the launch zone. In addition, compared with other nodes at the same level, the risk probability of the process control is excessively high; therefore, the control of the rocket launch process should be made stricter. Furthermore, the risk of the rocket stability is also high, and the rocket stability should thus be further enhanced. Although the casualty risk probability is low, it should be strictly controlled due to its high severity. The rocket delay risk can be considered after optimising the other risk events.

**Table 8.** Risk probability of the nodes in the BN.

| Node | Risk Probability |
| --- | --- |
| Risk in space launch | 0.0306 |
| Risk in technical zone | 0.0286 |
| Risk in launch zone | 0.030 |
| Risk in measure-control system | 0.0244 |
| Process control risk | 0.0518 |
| Rocket stability risk | 0.0318 |
| Launch delay risk | 0.0213 |
| Casualty risk | 0.0213 |

The fuzzy posterior probabilities are also obtained in the posterior reasoning. According to the different states of the root node "Launch site risk" as "No evidence", "High risk", "Medium risk", and "Low risk", respectively, the fuzzy posterior probabilities of each node can be obtained, as shown in Figure 7. The posterior probabilities indicate that the probability states with "No evidence" is between the "Low risk" state and the "Medium risk" state, which not only reflects the relatively high success rate of space launch tasks, but also reflects the non-negligible risk in the process of space launch. Moreover, as the parent nodes of "Risk in launch zone", the node "Process control risk" and the node "Rocket stability risk" are most affected by "Launch is high risk" = "High risk", which further proves that "Process control risk" should be paid enough attention in the risk management of space launch system.

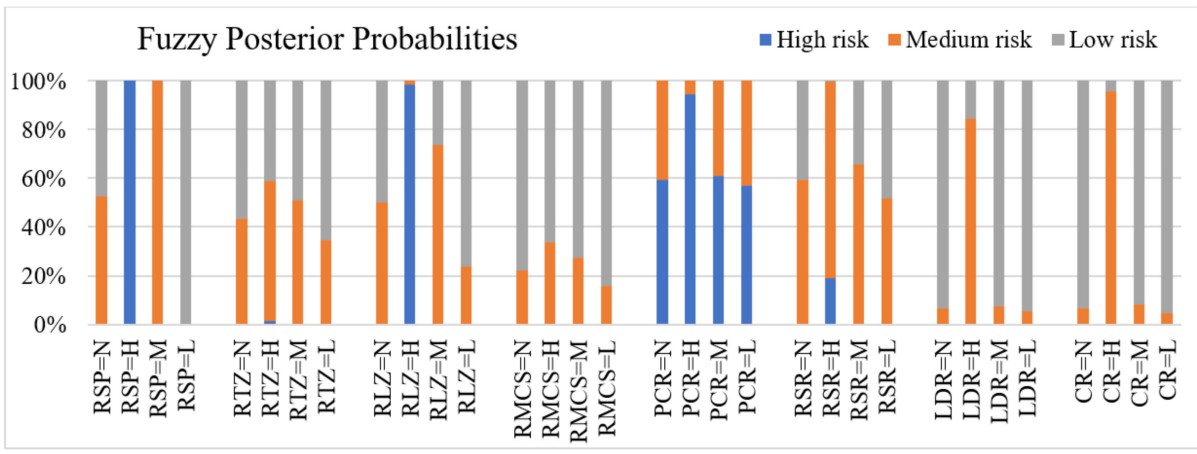

**Figure 7.** The fuzzy posterior probability diagram of the node "Risk in space launch".

*4.2. Risk Path Analysis*

Assuming that the SLS is in a high-risk state, that is, by setting the probability of the node "Risk in space launch system" in the high-risk state as 100%, the reverse reasoning based on the BN is used to determine the main reasons leading to the high risk pertaining to the SLS and to analyse the risk conduction path. The results of the reverse reasoning are shown in Figure 8. These results indicated that if the SLS is in a high-risk state, its parent node "Risk in launch zone" is most likely to be in a high-risk state with a probability of 98%; the parent node "Risk in technical zone" is most likely to be in a medium-risk state with a probability of 57%, and parent node "Risk in measure-control system" is most likely to be in a low-risk state with a probability of 66%. Therefore, the high risk of the SLS is mainly caused by the risk in the launch zone, followed by the impact of the risk in the technical zone.

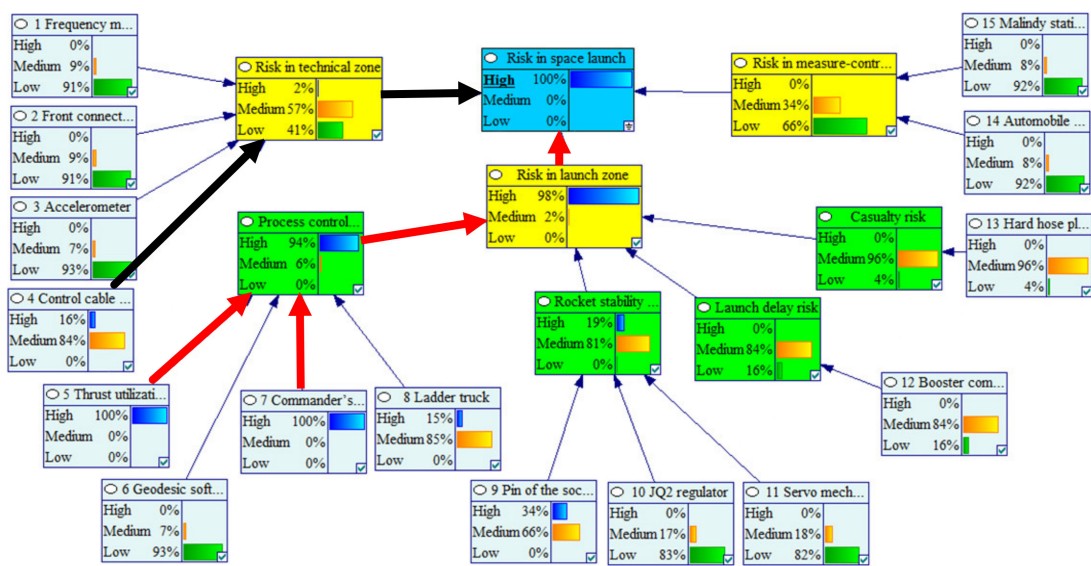

**Figure 8.** Reverse reasoning results of the BN.

Using this method, the risk causes are further traced. For the node "Risk in launch zone", the risk probability of node "Process control risk" is the highest in the parent node; when traced back to the parent node of "Process control risk", the risk probability of the nodes "5 Pre-computer" and "7 Commander's command" is the highest in the parent node. Therefore, to effectively reduce the risk pertaining to the SLS, the relevant personnel should focus on the risk events "5 Thrust utilization computer" and "7 Commander's instruction",

and strive to reduce the risk probability by optimising the launch process or strengthening the personnel supervision.

On the basis of the above methods, the main and secondary risk conduction paths could be determined by retrospective analysis of the risk causes of the SLS. The main conduction path is as follows: "5 Thrust utilization computer" or "7 Commander's instruction" → "Process control risk" → "Risk in launch zone" → "Risk in space launch". The secondary conduction path is as follows: "4 Control cable socket" → "Risk in technical zone" → "Risk in space launch", as shown in Figure 8 (the bold red arrows indicate the main conduction path, and the bold black arrows indicate the secondary conduction path).

### 4.3. Sensitivity Analysis

The sensitivity analysis of the system risk by using the BN can help the analysts determine the risk events that have the greatest impact on the system risk, to provide a more accurate event probability or conditional probability of these risk events when optimising the SLS. The sensitivity analysis result of the risk in the SLS is shown in Figure 9, the depth of the red colour indicates the sensitivity of the node to the node "risk in space launch system", which means the redder the node is, the more sensitive the node is to "risk in space launch. As shown in Figure 9, the probabilistic changes in the nodes "Rocket stability risk" and "13 Hard hose plug for filling" have the greatest impact on the risk of the total system. The sensitivity ranking of each node to the risks in the SLS is as shown in Figure 10, which indicates that the change in the risk in the technology zone, launch zone, and measure-control system has the greatest impact on the total system risk. The colour of the bar shows the direction of the change in the state of "risk in space launch system", red expresses negative and green positive change. In addition, the probabilistic changes in the key risk events 13, 14, and 15 have a greater impact on the total system risk. Considering the above analysis results, in the future risk assessment of SLS, the probability acquisition accuracy of the key risk events 13, 14, and 15 should be improved to reduce the impact of the probability estimation accuracy on the system risk assessment.

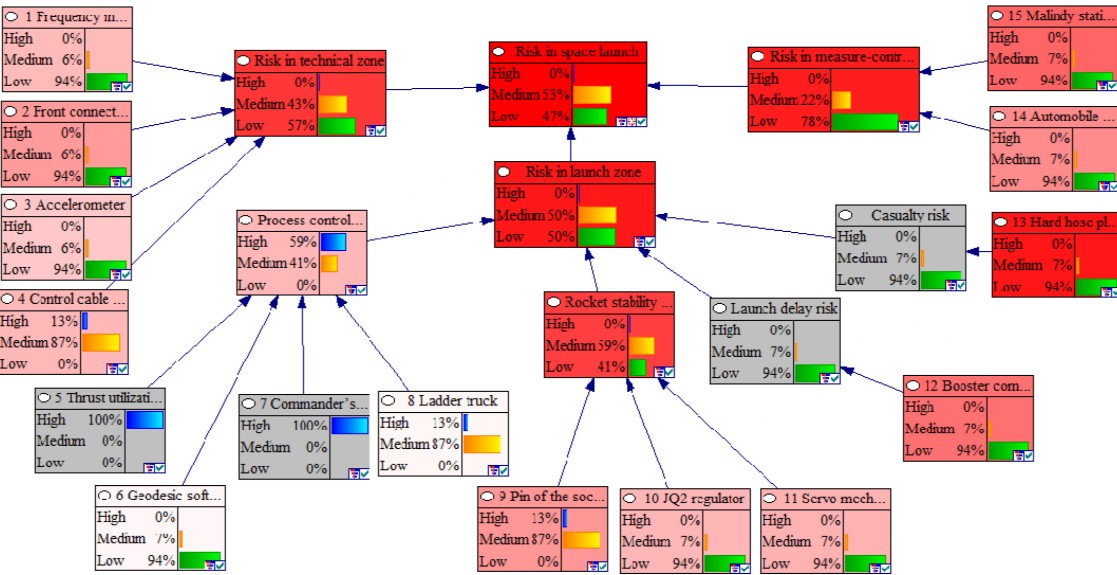

**Figure 9.** Sensitivity analysis results for the risk in a space launch.

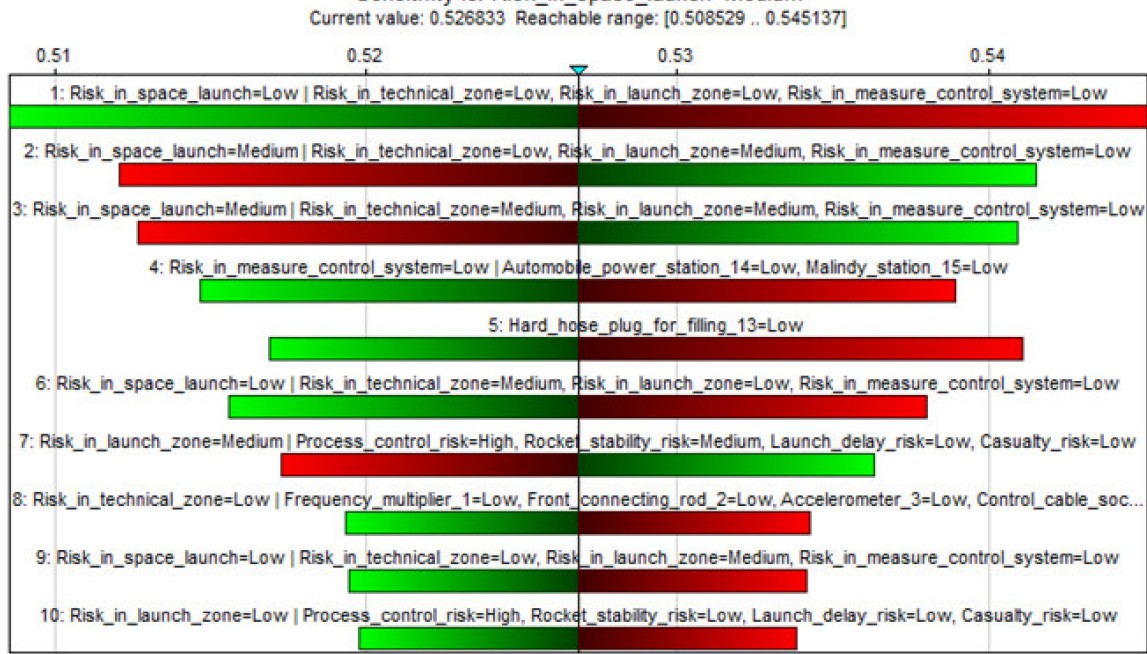

**Figure 10.** Sensitivity ranking for the medium risk in a space launch.

*4.4. Discussion*

In the above case, the fuzzy theory and expert experience are used to evaluate the risk of SLS comprehensively, and historical statistics are used reasonably, which reduces the impact of uncertainty. Some discussion is provided concerning the methods used in this paper.

(1) Fuzzy method

The fuzzy membership function is used for probability transformation, and the fuzzy score for risk level is used to determine the CPT. The proposed method is more accurate and reasonable than direct fuzzy evaluation of CPT. Furthermore, as the main uncertainty source of the BN, the CPTs are constructed with the aid of fuzzification in this paper. To reduce the uncertainties, the traditional CPT construction method exerts certain requirements on the workload of the expert evaluation, and thus, a one-sidedness problem exists when the experts evaluate a large number of conditional probabilities, which makes the construction of the CPTs time-consuming, laborious, and inaccurate. In view of the shortcomings of the traditional CPTs construction method, this paper appropriately fuzzified the risk probability of nodes and fully considered the influence weight of each parent node on the child node to obtain more objective CPTs.

(2) Case of SLS

Through the field investigation and data collection, the authors sorted out some krisk events that may lead to accidents, and obtained 15 key risk events and their frequency. After establishing the application framework and BN method, domain experts are invited to evaluate, and a complete BN model is finally obtained. Through Bayesian analysis, these results guide the engineering practice to reduce the occurrence of risk events and the probabilities of rare events are also fully considered. Due to the limitations of the proposed method, the accuracy of expert evaluation, and the size of data samples, the results obtained cannot fully reflect the actual situation. To solve this problem, in addition to optimising the method and improving the status of data acquisition, BN can be updated by using the posterior reasoning of BN to be more in line with the actual situation.

Generally speaking, the application framework and the method of building CPT are well used in the case of SLS. This framework makes the construction process and effect of BN clear, while the construction method of CPT can reasonably obtain CPT by using expert experience and fuzzy method.

## 5. Conclusions

The launch process of aerospace engineering is a stage in which equipment failures and human errors occur frequently, and many catastrophic risk events occur in this stage. PRA plays an important role in analysing and reducing risk. BN is a powerful risk assessment tool, and fuzzy method helps to reduce the uncertainty caused by the subjective evaluation of experts. This paper establishes a PRA framework for SLS using BNs with fuzzy theory and discusses the uncertainties in BNs. The framework summarises the BN application method from three layers of principle, model, and result. The framework clearly reflects the principles used in the construction of the BN model, and also show the effect and relationship of various BN analysis results.

Compared with previous studies, and based on the BN model, the contribution of this paper can be summarised as follows: (1) fifteen key risk events during the rocket launch were analysed and divided into risk in technical zone, risk in launch zone, and risk in ground measure-control system by experts, with the structure of BN constructed based on this; (2) a quantitative risk assessment analysis model was constructed for the rocket launch process, an improved CPT construction method was proposed based on the fuzzy theory, which can solve the problem of excessive reliance of the traditional CPT construction method on the expert evaluation; (3) risk path inference and sensitivity analysis of SLS were conducted. Our approach is well demonstrated in the case study, which help decision makers to make optimal resource allocations with limited resources.

**Author Contributions:** Conceptualization, X.P. and W.Z.; methodology, S.D.; software, T.L.; validation, T.L., S.D. and L.W. (Liqin Wang); formal analysis, L.W. (Lijing Wang); investigation, T.L.; resources, W.Z.; data curation, L.W. (Liqin Wang); writing—original draft preparation, S.D.; writing—review and editing, S.D.; visualization, X.P.; supervision, X.P.; project administration, X.P.; funding acquisition, W.Z. All authors have read and agreed to the published version of the manuscript.

**Funding:** This research was funded by National Natural Science Foundation of China grant No.72071011.

**Data Availability Statement:** Not applicable.

**Conflicts of Interest:** The authors declare no conflict of interest.

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
