# Peer review of "Probabilistic Risk Assessment in Space Launches Using Bayesian Network with Fuzzy Method"

_aerospace, doi:10.3390/aerospace9060311_

Round 1
Reviewer 1 Report
1. A large part of it depends on subjective opinion of the experts and there must have been enough number of experts for the same process to improve the defects that might not be fully scientific.
In the end, the relation between the parent nodes and child nodes also seems to be determined by experts.
Sensitivity analysis as well, it seems to be just a result of experts' evaluation.
Considering the concerns aforementioned, the manuscript would look better if it is clarified on how many experts were involved in the study.
2. Generally speaking, input variables have a coupling effect between them, which should be considered for better results.
3. Figure 7 is not readable properly in the black and white version.
4. Do figures 8,9 and 10 have any intention for their data using their colour? If there are, those should be clarified. Figure 10 is also not much distinctive in the black and white version.
5. It should be double checked whether all abbreviations used are defined before their first use. For example, CPT doesn't seem to be defined before its use.
6. The Bayesian Network used in this study was evaluated favourly without clear evidence. This should be supported by data.
7. The word 'America' appears several times, which seems to mention the US, but it might be confusing to potential readers as it also means South America and Central America as well.
8. Introduction Line 51-63, Page 2
There are some references mentioned in the paragraph, but the logical flow is unclear. It is just an arrangement of some pieces of work without any clear meaning. Further details for each reference should be added to make clear what were previous achievements, what were their limitations, and what is the novelty of this work.
9. Introduction, Line 70, Page 2, "A BN is suitable for the.."
Bayesian Network should be first introduced with its definition before its characteristics. What is the definition Baysian Network, why they were used in different applications, what were the achievements of previous studies using BN with more details should be addressed.
10. References regarding the CPT should be also added to the Introduction.
11. Line 244, Page 7, "Many types of launch vehicles in Russia, the United States, Europe, Japan, China and other countries have adopted liquid hydrogen and oxygen as propellants"
This statement should be supported by reliable references.
Reviewer 2 Report
This work is devoted to the application of risk assessment methods to the launch of launch vehicles. Well-known methods of Bayesian networks and fuzzy logic are used. The application of these methods in this problem allowed us to identify and rank the most critical aspects of the launch vehicle launch and, moreover, the authors identified the most critical elements of the conditional probability table. The method allows you to identify the causes of risk at different levels. I think these are interesting results. The article is written in a good and understandable language. I have a few small comments:
1) Please provide in the introduction references to the literature on Bayesian networks and fuzzy logic that is good from your point of view and where they work together
2) The paper does not define the abbreviation CPT anywhere, please correct this
3) Wherever summation is performed, the summation variable should be noted, for example in formulas 6-9.
4) In Conclusion, it should be noted not just what has been done (now the Conclusion is more like an Abstract), but specific conclusions from the study. For example, that the most important and critical aspects of the launch have been identified (and which aspects exactly!), the most critical elements of the conditional probability table (and what elements exactly) and so on.
I recommend minor revision of the paper.
Round 2
Reviewer 1 Report
The revised manuscript looks much better than the previous version. Most of the questions and comments raised by this reviewer have been answered and/or reflected. However, it would make it look even better, if the authors could properly address previous comment #9, "Bayesian Network should be first introduced with its definition before its characteristics. What is the definition Baysian Network, why they were used in different applications, what were the achievements of previous studies using BN with more details should be addressed.", as still there is no 'definition' of it. The introduction related to that part should start with 'what' is the Baysian Network and 'what' is Fuzzy Method, rather than what they can do, how they are considered, and how they're used. Better to start with its 'definition' rather than 'characteristics'..
